# Lignin Nanoparticles Produced from Wheat Straw Black Liquor Using γ-Valerolactone

**DOI:** 10.3390/polym16010049

**Published:** 2023-12-22

**Authors:** Lianjie Zhao, Yingchao Wang, Qiang Wang, Shanshan Liu, Xingxiang Ji

**Affiliations:** State Key Laboratory of Biobased Material and Green Papermaking, Qilu University of Technology (Shandong Academy of Sciences), Jinan 250353, China; zlj980701zlj@163.com (L.Z.); wyc19940530@126.com (Y.W.)

**Keywords:** wheat straw, black liquor, lignin nanoparticle, γ-valerolactone

## Abstract

The valorization of the black liquor produced during the chemical pulping of wheat straw is the key to the sustainable use of this abundant agricultural waste. However, the silica problem has hampered the recovery process. Herein, nanoprecipitation technology was used to produce lignin nanoparticles (LNPs) from wheat straw black liquor using γ-valerolactone (GVL) as a solvent and water as an anti-solvent. The results showed that a uniform, well-dispersed, and stable LNP was produced. The particle size and Zeta potential of 161 nm and −24 mV of the LNP suspension were obtained at a GVL concentration of 87%. The chemical structure and bonding of the lignin were adequately preserved after nanoprecipitation based on two-dimensional heteronuclear single quantum coherence nuclear magnetic resonance (2D-HSQC NMR) spectroscopy, Fourier transform infrared (FTIR) analysis, and thermal stability was improved based on thermogravimetric analysis. In addition, the abundant phenolic hydroxyl groups of LNP quantified by ^31^P-NMR analysis are beneficial for chemical cross-linking and modification. This work not only achieved the valorization of wheat straw black liquor but also opened up a new avenue for advanced nanomaterials.

## 1. Introduction

Facing the scarce depletion of fossil reserves and environmental pollution, more and more attention is being placed on finding sustainable alternatives to replace fossil resources [1]. Currently, agricultural waste is considered as a promising solution due to its abundant availability, renewability, and potential to produce energy, fuel, and materials based on the biorefinery concept [2]. Wheat straw, a ubiquitous agricultural residue, reaches an annual global output of approximately 0.887 billion tons [3]. Through the chemical pulping process, the primary components of wheat straw (e.g., cellulose, lignin, and hemicellulose) can be effectively fractionated, unlocking opportunities for the production of paper, energy, and materials [4]. This underlines the key role of agricultural waste, particularly wheat straw, in contributing to sustainable resource utilization and reducing environmental impact.

However, during the pulping process, the high content of silica in wheat straw can also dissolve into black liquor, which is a by-product of chemical pulping and contains lignin, hemicellulose, and other organic compounds [5]. Currently, black liquor is valorized by recovering chemicals and organic compounds, such as tall oil and lignin (together with alkali recovery and energy production) [6]. Yet, the presence of silica with high levels caused serious problems, such as hindering the settling of the lime mud during recausticisation and adversely affecting the clarity of white liquor while also reducing the possibility of valorizing the side streams [7].

Despite extensive research efforts to address these complications, including precipitating silica onto pulp as insoluble silicates, using CO_2_ or combustion flue gases to form an insoluble silicic acid gel and pre-desiliconization, not one has offered a comprehensive solution to the silica-related problems in order to valorize the side streams [8]. As a result, the widespread use of wheat straw in the pulping industry is severely limited, highlighting the critical need for innovative solutions that can effectively reduce the adverse effects of silica, thereby unlocking the full potential of wheat straw as a sustainable resource in the pulping process.

Thanks to nanotechnology, it offers an alternative solution to the challenges posed by the high silica content in black liquor derived from the wheat straw pulping process through the production of lignin nanoparticles (LNPs) [9,10]. LNPs exhibit enhanced reactivity due to their large specific surface area and increased surface energy, which facilitates efficient binding with other atoms [11]. The inherent properties of lignin, including biodegradability, antibacterial activity, antioxidation, and UV absorption, contribute to the superior thermal stability, UV resistance, and non-cytotoxicity of LNPs [12,13,14]. After lignin precipitation, black liquor could be reused in the pulping process, forming a closed-loop process. In addition, the production of LNPs has the advantage of not only allowing the high-value utilization of lignin but also reducing the potential safety risks associated with the biological hazards of conventional nanomaterials [15]. Current research has explored various methods for LNP production, which can be categorized into nanoprecipitation, mechanical, and biological methods [16]. Among these, the nanoprecipitation method stands out for its high efficiency and promising potential in large-scale applications. This method exploits the amphiphilicity of lignin molecules, where hydrophilic functional groups such as carboxyl and hydroxyl, together with the hydrophobic structure of the benzene ring backbone, self-assemble into spherical particles using a lignin solvent and anti-solvent system [17]. Despite the efficiency of nanoprecipitation, a critical consideration in LNP production is the choice of the solvent. Commonly used solvents such as tetrahydrofuran, acetone, and ethanol, although effective, are volatile, flammable, and sometimes toxic [18]. Addressing this issue is critical to ensure the safety and scalability of LNP production methods. Research efforts to develop sustainable and environmentally benign solvents are ongoing. Addressing this aspect is crucial for ensuring the safety and scalability of LNP production methods. Research efforts aimed at developing sustainable and environmentally benign solvents are ongoing, promising further advancements in the large-scale application of lignin nanoparticles and contributing to both the sustainable utilization of agricultural waste and the evolution of green nanotechnologies.

γ-Valerolactone (GVL), a novel biomass-derived solvent, represents an ideal solution for lignin’s dissolution due to its green, efficient, environmentally friendly, and sustainable properties. Obtained from biomass via dehydration and hydrogenation [19], GVL is a versatile organic solvent known for its miscibility with water and has applications in food, fuel additives, and polymer monomer synthesis [20,21]. The advantages of GVL, including low vapor pressure, a high boiling point, room temperature stability, low toxicity, and sustainability, all contribute to its excellent solvent properties, making it a promising and environmentally friendly choice for various industrial applications [22,23]. In recent years, it has been used in biomass pretreatment, the non-enzymatic saccharification of biomass, and the dehydration of hemicellulose to produce furfural [24,25]. For example, Gürbüz et al. [26] found that the hemicellulose fraction of lignocellulose was converted to furfural with a high yield of 80% in a single-phase system using GVL as a solvent and the acid solid H-mordenite as a catalyst; the decomposition reactions of the product can be significantly reduced by minimizing the water concentration in GVL. In 2016, Kong et al. [27] also indicated that the sugarcane bagasse, hydrolyzed in a mixed GVL/water solution, can be an alternative substrate for the low-cost production of biobutanol. Furthermore, many studies have shown that the binary solvent system composed of GVL and a co-solvent (such as water, ionic liquid, etc.) can effectively dissolve different types of lignin [28,29]. For example, Xue and co-workers [30] used the binary solvent systems consisting of GVL and a cosolvent (such as DMSO; DMF; ionic liquids 1-butyl-3-methylimidazolium acetate ([Bmim]OAc); and 1-allyl-3-methylimidazolium chloride ([Amim]Cl)) as rapid systems for dissolving different types of lignin. It was found that the cosolvent content in GVL-based systems significantly affects the solubility of lignin. More importantly, they concluded for the first time that the solubility of lignin is associated with the hydrogen bonding alkalinity parameter of the solvent, β, the relationship between the values depending on both the solvent and the lignin, which clarifies the current controversy. To date, there are few studies on the production of LNP from wheat straw black liquor using GVL.

In this study, γ-valerolactone (GVL) was used as a key component in the production of lignin nanoparticles (LNPs) derived from wheat straw black liquor, employing nanoprecipitation technology with varying GVL-to-water ratios. The resulting LNPs were subjected to a comprehensive characterization to elucidate their morphological, physical, and chemical attributes. Scanning electron microscopy and Nano Zetasizer analysis provided insights into the morphological features and size distribution of the as-prepared LNPs. A multi-faceted approach, including a two-dimensional heteronuclear single quantum coherence nuclear magnetic resonance (2D-HSQC NMR), Fourier transform infrared (FTIR), and quantitative ^31^P-NMR spectroscopy, facilitated an in-depth examination of the chemical structure and bonds within the LNPs. Additionally, the thermal stability of the LNPs was scrutinized using a thermogravimetric analyzer. This investigation not only facilitated the valorization of wheat straw by converting black liquor into LNPs but also drove the development of advanced nanomaterials characterized by enhanced performance and reduced environmental impact, opening new avenues for sustainable and innovative applications.

## 2. Experimental Section

### 2.1. Materials

Wheat straw was kindly provided by a pulp mill located in eastern China. The cellulose, hemicellulose, lignin, ash, and silica contents of the wheat straw were 42.5%, 26.8%, 17.3%, 7.04%, and 5.43%, respectively. Hydrochloric acid (HCl) was purchased from Sinopharm Chemical Reagent Co., Ltd. (Shanghai, China). γ-Valerolactone (GVL) was supplied by Shanghai Macklin Biotechnology Co., Ltd. (Shanghai, China). 4-nitroaninline (4NA) and N, N-diethyl-4-nitroaniline (DENA) were obtained from Sinopharm Chemical Reagent Co., Ltd. (Tianjin, China). Reichardt’s dye, cyclohexanol, chromium (III) acetylacetonate, deuterated dimethyl sulfoxide ([D_6_]DMSO), anhydrous pyridine, deuterated chloroform (CDCl_3_), and 2-chloro-4,4,5,5-tetramethyl-1,3,2-dioxaphospholane (TMDP) were all purchased from the Sigma-Aldrich company (Shanghai, China). All chemicals were of an analytical grade and used without further purification. Deionized water was used in all the experiments.

### 2.2. Recovery of Lignin

The soda cooking process was chosen for digestion, with a NaOH dosage of 18%, a maximum cooking temperature of 160 °C, a cooking time of 1 h, and a liquid/solid ratio of 5:1. An electronically controlled digester (15 L) was used for cooking with a heating rate of 2 °C/min. After cooking, the black liquor was collected from the pulp using a filter bag. Later, the lignin was precipitated by acidifying the black liquor with HCl to adjust the pH to 2. The precipitate was washed with deionized water and then centrifuged at 4000 rpm for 0.5 h and finally vacuum-dried at 40 °C for 3 days to obtain the lignin. The detailed production procedure was based on previous research [31]. The pulp yield, viscosity, kappa number (Kappa No.), beating degree, breaking length, tear index, and burst index were determined according to the Technical Association of the Pulp and Paper Industry (TAPPI) method. Specifically, the α-cellulose content was determined according to TAPPI T 203 cm-09. Viscosity was determined according to TAPPI T 230 om-94 in a cupriethylene diamine (CED) solution at a 0.5% cellulose concentration. The intrinsic viscosity was then calculated according to the previous study [32]. The beating degree was measured following the TAPPI T 248 method through the PFI-refiner. The breaking length, tearing index, and bursting index of the pulp were analyzed based on TAPPI T 404 wd-03, T 414 om-04, and T 403 om-02, respectively. All tests were performed in triplicate, and the average value was reported. Prior to testing, the sample was air-dried for 48 h at 25 °C and 50% RH.

### 2.3. Preparation of Lignin Nanoparticle

Lignin nanoparticles (LNPs) were prepared via nanoprecipitation using GVL as the solvent and DI water as the antisolvent based on our previous study [25]. Specifically, 1 g of lignin was dissolved in 100 mL of the GVL/water binary solvent (GVL concentration was set to 30%, 42%, 65%, and 87%, respectively), followed by ultrasonication for 20 min. After centrifugation at 10,000 rpm for 20 min, 90 mL of water was slowly injected into the saturated lignin solution (10 mL). The suspension was then centrifuged at 10,000 rpm for 0.5 h to precipitate the lignin nanoparticles. The resulting LNPs were redispersed in DI water and then centrifuged again to remove the GVL solvent. The whole process was repeated five times to obtain a well-dispersed LNP.

### 2.4. Morphology Characterization

The scanning electron microscope (SEM, Hitachi SU-70, Tokyo, Japan) was used to observe the surface morphology of LNP at 5 kV of acceleration voltage. Prior to this observation, the LNP dispersion (0.005%) was ultrasonically treated for 10 min and then dropped onto a grid covered with carbon membrane and air-dried [33].

### 2.5. Particle Size Distribution

The particle size distribution of the lignin nanoparticle was tested using a Malvin Zetasizer Nano ZS90 analyzer (Malvern, UK) at 25 °C. The LNP dispersion (0.01%) was first sonicated for 10 min and then poured into a quartz cuvette with four clear sides for measurement. The analysis was executed in triple, and the results were averaged.

### 2.6. NMR Analysis

Quantitative ^31^P-NMR analysis was conducted via a nuclear magnetic resonance (NMR) spectrometer (400 MHz AVANCE III, Karlsruhe, Germany) with a Quad probe dedicated to ^31^P, which was carried out according to the modified method reported in the literature [34]. The sample preparation was firstly weighed as 20 mg of the dried LNP sample and then dissolved into 0.5 mL of the mixture of anhydrous pyridine/CDCl_3_ (1.6:1, *v*/*v*), which included the relaxation additive of chromium (III) acetylacetonate (3.6 mg/mL) and the internal standard of cyclohexanol (4 mg/mL). Afterward, the hydroxy groups of lignin were phosphorylated with 0.1 mL of TMDP for 15 min. Once completed, the as-prepared sample was transferred into a nuclear magnetic tube for testing.

The 2D-HSQC NMR analysis was carried out on a nuclear magnetic resonance (NMR) spectrometer (Tokyo, Japan). Similarly, the sample preparation was weighed as 80 mg of freeze-dried LNP powder and then dissolved into 0.5 mL of [D_6_]DMSO, and centrifuged to remove the insoluble material after sonication; finally, the supernatant was detected using a nuclear magnetic spectrometer (Tokyo, Japan). The following adjustments were performed using a standard Bruker pulse sequence following the previous literature [17], i.e., a 13 ppm spectra width in the F2 (^1^H) dimension with 1024 data points (95.9 ms acquisition time) and a 210 ppm spectra width in an F1 (^13^C) dimension with 256 data points (6.1 ms acquisition time). A 90 ° pulse, 0.11 s acquisition time, 1.5 s pulse delay, and 48 scans were followed to acquire HSQC spectra. The ^1^J_C–H_ of 145 Hz was used in this experiment. Data processing was executed using Bruker BioSpin TopSpin 4.1.1 software and Mestre Nova 14 software.

### 2.7. FTIR Analysis

Fourier transform infrared (FTIR) spectra were recorded by an ALPHA FTIR spectrometer (Rosenheim, Germany) using a standard ATR probe over the wavenumber range of 4000–600 cm^−1^ with a resolution of 4 cm^−1^ for 32 scans.

### 2.8. Thermal Stability

The thermal stabilities of lignin and LNP were determined using a thermogravimetric analyzer (New Castle, DE, USA) under a N_2_ atmosphere at a 50 mL/min flow speed. In this test, about 15 mg of freeze-dried LNP powder was placed in an alumina crucible and then heated from 25 to 600 °C at a heating speed of 10 °C/min.

## 3. Results and Discussion

### 3.1. Valorization of Wheat Straw

Wheat straw, an abundant agricultural residue, contains cellulose, hemicellulose, and lignin compositions of 42.5%, 26.8%, and 17.3%, respectively (Figure 1). A relatively high pulp yield of 41.5% could be obtained under the pulping conditions. The intrinsic viscosity and Kappa No. were within a reasonable range. After mechanical refining and papermaking, a rather strong paper was produced with a breaking length of 7.05 km, a tearing index of 4.69 mN·m^2^/g, and a bursting index of 4.76 kPa·m^2^/g at a beating degree of 45 °SR. The properties of the resulting pulp were comparable to those of other pulps [35,36]. Therefore, it has a rather wide range of applications, such as packaging, printing, newspapers, and paperboard. The most important and detrimental problem of such pulping is the presence of high silica content in wheat straw, which is about ten times higher than wood. The high level of the silica content caused serious problems (known as the “silica problem”), such as inhibiting the settling of the lime sludge during causticization and adversely affecting the clarity of the white liquor during alkali recovery sequences. These challenges lead to operational complexity and increased chemical and energy consumption. The integration of nanoprecipitation technology into the pulping process is emerging as an alternative solution to effectively reduce silica-related issues. This integration promises streamlined operations, reduced energy consumption, and increased efficiency, providing a sustainable approach to wheat straw pulping.

The production of lignin nanoparticles (LNPs) via the self-assembly method depends on the solubility contrast of the lignin in solvents. As water-insoluble lignin undergoes a transformation process into a colloidal dispersion, the resulting LNP exhibits remarkable antioxidant activity and inherent chemical modifiability. Its unique properties, including biodegradability, antibacterial efficacy, and UV absorption capability, make LNPs applicable for diverse applications, such as drug delivery, antibacterial agents, and emulsion stabilization [37,38]. The injection of lignin from a good solvent to an anti-solvent induces the formation of LNP, where exposed hydrophilic functional groups and shielded hydrophobic moieties culminate in a water-dispersible colloid. Furthermore, the inherent bio-based recyclability of γ-valerolactone (GVL) enhances the biocompatibility of LNP, expanding its potential utility in Pickering emulsions, composite fillers, drug carriers, and more. This study not only addresses the challenges associated with black liquor recovery and silica in papermaking but also provides valuable insights into the sustainable use of wheat straw. The versatile properties of LNP, coupled with its environmentally friendly production process, pave the way for its integration into various industrial sectors, demonstrating the potential of lignocellulosic biomass in the advancement of green and sustainable technologies.

### 3.2. Morphology and Particle Size Distribution of LNP

To observe the microstructure, the morphology of the LNP under different GVL concentrations is shown in Figure 2a–h. As can be seen, all of the as-prepared LNPs had an adequate spherical shape, indicating the good solubility property of GVL for wheat straw lignin and the success of the dropping nanoprecipitation process [39]. Also, the uniformity and dispersion of LNP gradually improved with the increasing GVL concentration (Figure 2i–l). For example, LNPs with an uneven particle size in the distribution range of 80–1700 nm and a high average size of 433 nm were produced at a GVL concentration of 30% (Figure 2i). When the GVL concentration was increased in the range of 42–87%, the particle distribution and size were narrowed and decreased gradually. The LNPs with a distribution range of 91–460 nm and a particle size of 161 nm occurred at a GVL concentration of 87% (Figure 2l). This can be explained by the Hildebrand solubility parameter theory, i.e., when the δ-value of the solvent is similar to that of lignin, it has better solubility for the lignin [40]. Moreover, Chen et al. [41] found that a high GVL concentration of 87% has better lignin solubility when producing lignin nanoparticles to enhance the antimicrobial activity of essential oils.

### 3.3. Dispersion Stability Analysis of LNPs

Long-lasting stability is essential for LNPs during their storage, transportation, and application. The variation in the hydrodynamic diameter and Zeta potential was monitored at different standing times and is shown in Figure 3. It was discovered from Figure 3a that the average particle size of LNPs gradually decreased with the increasing GVL concentration, which is in agreement with the SEM results (Figure 2). The average particle size was adequately maintained for 30 days (Figure 3a,b). The hydrodynamic diameters of LNP obtained under different GVL concentrations changed slightly from 433 to 450 nm, from 375 to 402 nm, from 217 to 242 nm, and from 161 to 186 nm, respectively, within 30 days, which demonstrated that LNP had good dispersion stability in water (Figure 3b). In addition, the absolute value of Zeta potential gradually increased with the increase in the GVL concentration and changed slightly over 30 days (Figure 3c,d). As shown in Figure 3e, no significant particle enlargement and precipitation were observed within 30 days. The LNP dispersion also showed a strong Tyndall effect (Figure 3f). The negative charge was attributed to the hydroxyl and carboxyl groups exposed on the outside of the LNP, forming a double electron layer. Due to the repulsion of the double electron layer, the LNP exhibited good dispersion stability in DI water [42]. In addition, the higher absolute value of ζ-potential can benefit the stability of LNP through stronger repulsive force to prevent aggregation. The above results indicate that LNP had excellent dispersion stability. The good dispersibility meant that LNP could be used in many advanced fields, such as a carrier or stabilizer in water [43,44].

### 3.4. NMR Analysis of LNPs

The chemical structure and functional groups were analyzed using 2D-HSQC NMR and quantitative ^31^P-NMR. The LNPs prepared from the GVL concentration of 87% showed a better performance compared to the others, so the following analysis was focused on this sample. As shown in Figure 4, the side chain regions (δC/δH 50~90/3.0~5.8) and aromatic regions (δC/δH 100~150/6.0~8.0) were well observed. In the side chain regions, both samples showed an obvious signal of methoxyl (δC/δH 56.10/3.75). The coupling peak of the C_α_–H_α_, C_β_–H_β_, and C_γ_–H_γ_ appeared at 72.4/4.92, 83.4/4.38 and 59.9/3.5 of δC/δH, respectively. The resonance peak at δC/δH 63.4/3.6 was the C_γ_–H_γ_ coupling peak of the full structure of phenyl coumarin (C, formed by α-O-4 and β-5) [45]. The presence of the solvent GVL formant was observed in the LNP sample, indicating the residual solvent. In the aromatic region, three correlations of C_2_–H_2_ (δC/δH 111.5/6.96 ppm), C_5_–H_5_ (δC/δH 114.9/6.76 ppm), and C_6_–H_6_ (δC/δH 119.9/6.74 ppm) were observed for the guaiacol units (G), and δC/δH 128.4/7.17 ppm was originated from the C_2,6_–H_2,6_ aromatic of the H unit [46]. The resonance peaks observed at 110/7.35 and 123.1/7.20 of δC/δH belonged to the coupling peaks of C_2_–H_2_ and C_6_–H_6_ of the ferulate ester, respectively. The signal of ferulic acid indicated that ferulic acid in lignin was connected with the hydroxyl group of polysaccharides by an ester bond and could form an ether bond with the β position of the lignin [47]. The resonance peak at 130.2/7.48 of δC/δH was the coupling peak of C_2,6_–H_2,6_ of the coumarate structure. Similar coupling peaks were shown in both side chain and aromatic regions, which demonstrated that the chemical structure of LNP was adequately maintained during the nanoprecipitation process. In addition, the phenolic hydroxyl groups of guaiacol-OH, syringyl-OH, and p-Hydroxyphenyl-OH were 3.4, 1.96, and 0.66 mmol/g from quantitative ^31^P-NMR, illustrating the potential for cross-linking and a functional modification (Figure 4).

### 3.5. FTIR Analysis and Thermal Stability of LNPs

The chemical groups and thermal properties of lignin and LNP were also analyzed, and the results are shown in Figure 5. As seen in Figure 5a, a broad absorption peak appeared at 3420 cm^−1^, which was the tensile absorption peak of the hydroxyl [48]. While 1700 cm^−1^, 1599 cm^−1^, 1508 cm^−1^, and 1458 cm^−1^ corresponded to the C=O stretching vibration, the aromatic skeleton vibration, C-C stretching of an aromatic skeleton, and C-H stretching of an aromatic skeleton, were the unique absorption peaks of the lignin (Figure 5b). The C-H vibration absorption peak of the methyl group at 1427 cm^−1^, 1264 cm^−1^ was attributed to the O-H vibration of the phenolic ether group in lignin, and the absorption peak at 1120 cm^−1^ was correlated with the aromatic C-H deformation vibration, and 1030 cm^−1^ was the OH stretching vibration of primary alcohol [49]. Similar absorption peaks in the range of 1800~900 cm^−1^ were detected for both samples, which suggested that the chemical bond structure of lignin was also well-preserved during nanoprecipitation.

In addition, the TG (thermogravimetry) and DTG (derivative thermogravimetry) curves of different lignin nanoparticles are shown in Figure 5c,d, respectively. According to the TG curve in Figure 5c, the thermal degradation procedure of the samples was mainly divided into three stages. The first stage was from room temperature to about 150 °C, and the slight weight loss was attributed to the evaporation of the water absorbed in the LNP. The second stage was the severe weight loss at 150~450 °C, which was caused by the breaking of the molecular chains. At this stage, the LNP was decomposed into small molecules and gaseous products. The third stage was higher than 450 °C, where the residues were further decomposed into gas and residual char [25]. In addition, as shown in Figure 5d, the main thermal degradation data, including the onset of the thermal degradation temperature, the maximum degradation temperature, and the residues of lignin, were 193.5 °C, 359.1 °C, and 43.6%, respectively. These values were increased by 31.7 °C, 31.4 °C, and 0.9% for LNP compared to the original lignin, indicating an improvement in the thermal stability of lignin nanoparticles.

## 4. Conclusions

In this work, lignin nanoparticles (LNPs) were obtained from wheat straw black liquor via nanoprecipitation technology using γ-valerolactone (GVL) as the solvent and deionized water as the anti-solvent. The results show that the obtained LNPs had a spherical shape and good dispersibility in the water, which was negligibly changed even after 30 days of storage. The particle size of 161 nm and a Zeta potential of −24 mV were observed for the LNP suspension when using a GVL concentration of 87%. Nuclear magnetic resonance and Fourier transform infrared analyses confirmed that the as-prepared LNP exhibited a typical chemical structure, indicating that the chemical bonding of the original lignin was well preserved after the nanoprecipitation process. The functional groups and improved thermal properties made LNP a promising application. This study not only addresses the silica challenges associated with wheat straw chemical pulping but also opens avenues for the value-added utilization of lignin.

## Figures and Tables

**Figure 1 polymers-16-00049-f001:**
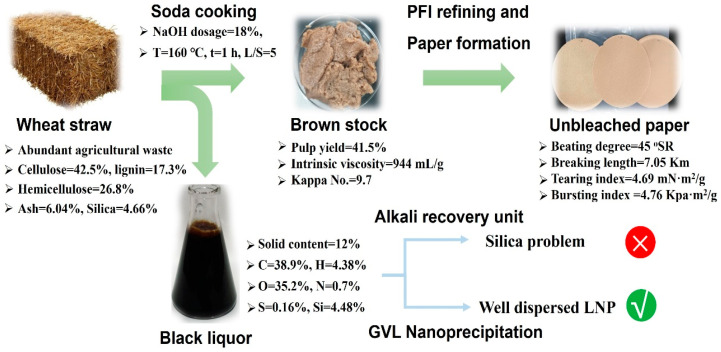
Integrating the nanoprecipitation of black liquor into the wheat straw pulping process.

**Figure 2 polymers-16-00049-f002:**
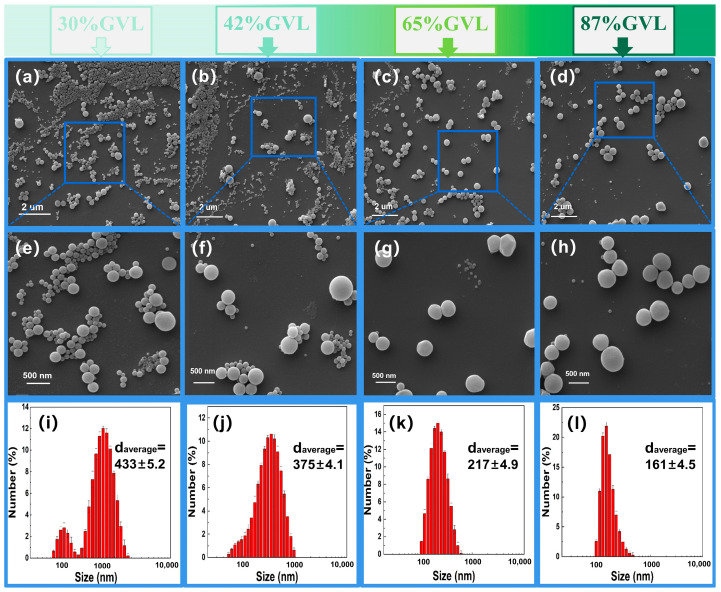
(**a**–**h**) Images of lignin nanoparticles obtained with different volume fractions of GVL at 10,000 and 40,000 magnifications observed by SEM; (**i**–**l**) Particle size distribution of lignin nanoparticles obtained with different volume fractions of GVL determined using a particle analyzer.

**Figure 3 polymers-16-00049-f003:**
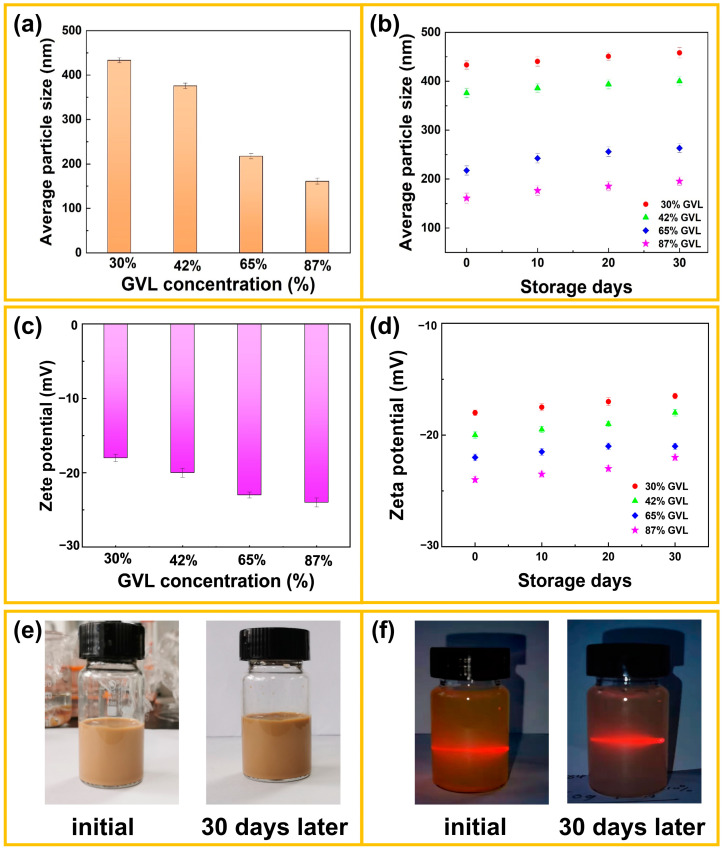
(**a**) The average particle size of LNPs under various GVL concentrations, (**b**) Average particle size of LNPs under various storage times, (**c**) Zeta potential of LNPs under different GVL concentrations, (**d**) Zeta potential of LNPs under different storage times, (**e**) Initial and stored 30-day LNP samples, (**f**) Tyndall effect of LNPs.

**Figure 4 polymers-16-00049-f004:**
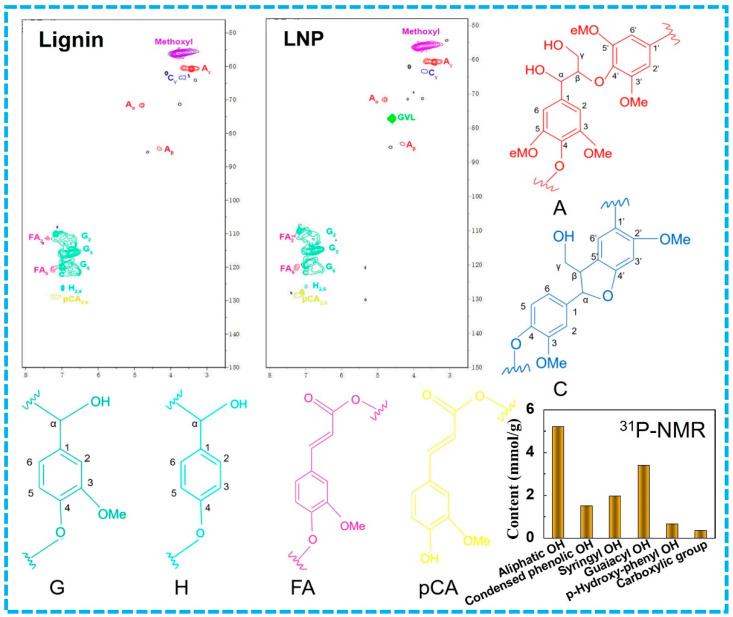
The 2D-HSQC NMR and ^31^P-NMR spectra of LNPs. (A: β-O-4, C: α-O-4, β-5, O-4, G: guaiacol units, H: p-hydroxyphenyl units, FA: ferulate ester, pCA: coumarate structure).

**Figure 5 polymers-16-00049-f005:**
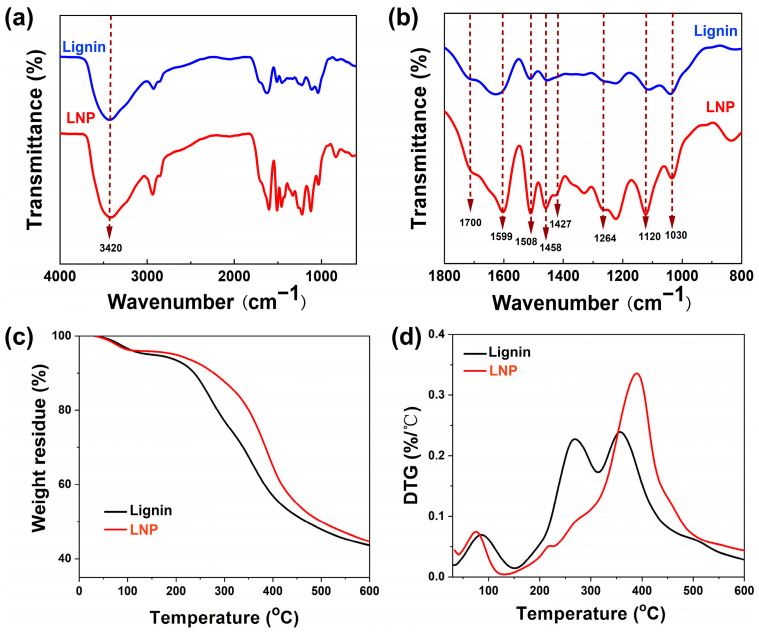
(**a**) FTIR spectra at 4000–600 cm^−1^, (**b**) the amplified FTIR spectra at 1800–800 cm^−1^, (**c**) TG, and (**d**) DTG curves of lignin and LNPs.

## Data Availability

The data required to reproduce these findings are available from the corresponding author upon request.

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
