# Peer review of "Lignin Nanoparticles Produced from Wheat Straw Black Liquor Using γ-Valerolactone"

_polymers, 2023, doi:10.3390/polym16010049_

Round 1

Reviewer 1 Report

Comments and Suggestions for Authors

This manuscript describes a study that can make a significant contribution to the valorization of wheat straw black liquor.

The methodology of the study is presented in detail. The results are clearly described. The article can be published, but after correction based on the following comments:

- Percentages should be written separately from the number. Check also throughout the text of the manuscript.

- The comma in line 93 is lost.

- Make sure that all the superscripts are written correctly. For example, for 31P, cm-1 and other. 

The degree of deuteration of pyridine is not specified. Or was deuterated pyridine used?

-  Line 185. The word "nanoparticles." in the title to Figure 1 is superfluous.

- The biggest questions are caused by the results of the SEM analysis. Based on the microphotographs presented in Fig. 2, it can be seen that in the case of 87 % GVL, nanoparticles are much smaller in number and they are large. Based on the presented 500 nm scale line, it can be seen that their diameter can reach 400 nm or more.  Whereas the authors note for this sample the average particle diameter of 161 nm, which are almost absent in the SEM image. Conversely, in the case of 30 % GVL, a large number of nanoparticles of a smaller diameter are observed, whereas the authors note an average diameter of 433 nm. Thus, these results look very contradictory.

Author Response

Please see the attached response letter.

Reviewer 2 Report

Comments and Suggestions for Authors

The current work looks at producing LNP from soda cooked wheat straw black liquor. There is no good justification for this effort other than the statement that a silica problem exists. The authors state that a silica problem exists without ever explaining what the silica problem actually is. The only justification for the work is that a silica problem exists.  The authors state that the LNP provides a potential economic product without justification of what that might be. (Also, in the form produced, there is no possibility that the material will have economic viability.)

The work is well done but why is it important. 

Ln 102 mentions TAPPI methods were used. Authors should list the specific standards which were employed in the work not just a generic statement as is currently in the text.

Comments on the Quality of English Language

The entire manuscript needs to be rewritten to improve the grammar.  Please have someone familiar with Standard English review and rewrite the manuscript.

Author Response

(The authors gave the same response as above.)

Reviewer 3 Report

Comments and Suggestions for Authors

In my opinion, the topic of the manuscript has the potential to fit within the scope of the journal. However, I advise major revision. The English sometimes is not so clear, so I advise to ask the help of a native-English speaker. Moreover, how the LNP resolves the problem of the silica is not explained in the text. Does the LNP remove the silica? Is it an alternative method to the typical recovery of alkali? Because the LNP removes silica, it is necessary to quantify the silica content before and after LNP precipitation. If it is an alternative method, the authors did not explain what to do with the black liquor after LNP precipitation.

Minor comments and examples of not fluent English.

Line 9: “The value…waste.” The sentence is not clear. The English is not fluent.

Line 22: I suggest adding “γ-valerolactone” as a Keyword.

Line 17: “while…analysis”. Probably, the authors mean “was analysed by thermogravimetric analysis.”. Right?

Line 18: “In addition…modification”. The English is not fluent. I advise “…phenolic hydroxyl groups of LNP quantified by 31P-NMR…”. Moreover, it is not clear to me how the phenols could benefit from the chemical crosslinking.

Line 30: remove the word “Fortunately”.

Line 35: “Traditionally…soda recovery.” I suggest adding that currently, black liquor is valorised by recovering chemicals such as Tall oil and lignin (together with alkali recovery and energy production). I advise adding a reference to support the statement.

Line 97: Were the Soda pulping and Lignin processes performed by the authors or from the pulping mill? If the authors performed them, I suggest splitting the 2.1 Section in two: 2.1 Materials with a list of what was used. 2.2 Recovery of lignin: where the precipitation lignin process is described.

Line 305: If acronyms are used like DTG, the first time they are reported, it is necessary to write the full name. Please check the entire test and all acronyms used.

Comments on the Quality of English Language

In some parts of the text, English is not clear/fluent. I suggest asking for the help of a native speaker. I reported some examples in the "comments and suggestions for authors".

Author Response

Dear reviewer:

Thank you for your review to our work. All the comments have been responsed  point-by-point. Please see attachment for further details. 

Sincerely,

Qiang Wang

Round 2

Reviewer 3 Report

Comments and Suggestions for Authors

The authors replied to my comments. The manuscript was improved. I suggest some minor revisions to clarify that this is an alternative valorization process.

Line 39: please change "Tall oil" with "tall oil".

Line 42: Based on the answer of the authors, "It is an alternative method to the typical recovery of alkali". I advise to modify the text.

I recommend writing from "...white liquor [7]." to "white liquor, reducing also the possibility of valorizing the side streams [7]."

Line 44: CO2

Line 45: I advise modifying the sentence "None has offered a comprehensive solution to the silica related problems [8]." to "...related, in order to valorize the side streams [8].".

Author Response

Dear reviewer:

Thanks for your kindly advise. It has been revised accordingly, please see the attachment.

Thanks

Qiang Wang
